# Unseen, untreated: chronic *Giardia lamblia* infections and the gaps in health systems

**Deiviane Aparecida Calegar¹, Filipe Anibal Carvalho-Costa²,³/⁺, Alda Maria Da-Cruz⁴, Maria Fantinatti⁴,⁵**

¹Ministério da Saúde, Secretaria de Vigilância em Saúde e Ambiente, Departamento de Doenças Transmissíveis, Brasília, DF, Brasil
²Fundação Oswaldo Cruz-Fiocruz, Instituto Oswaldo Cruz, Laboratório de Doenças Parasitárias, Rio de Janeiro, RJ, Brasil
³Universidade do Estado do Rio de Janeiro, Faculdade de Ciências Médicas, Departamento de Medicina Interna, Rio de Janeiro, RJ, Brasil
⁴Fundação Oswaldo Cruz-Fiocruz, Instituto Oswaldo Cruz, Laboratório Interdisciplinar de Pesquisas Médicas, Rio de Janeiro, RJ, Brasil
⁵Universidade do Estado do Rio de Janeiro, Faculdade de Ciências Médicas, Departamento de Microbiologia, Imunologia e Parasitologia, Rio de Janeiro, RJ, Brasil

Giardiasis is a chronic disease that impairs intestinal absorption and, independently of diarrhoea, contributes to growth and developmental delays, making it an important risk factor for protein-energy malnutrition in children. Its high prevalence is associated with poor sanitation and multiple faecal-oral transmission routes, including person-to-person spread, contamination of water and food, and zoonotic cycles. *Giardia lamblia* closely interacts with the intestinal epithelium, triggering inflammatory and immune responses. Despite its impact, giardiasis is underdiagnosed, and there is an urgent need for microscopy-independent diagnostic tools suitable for large-scale use. Treatment options include metronidazole or single-dose regimens such as secnidazole and tinidazole, which are more effective but not yet implemented at the community level. Because exposure is continuous, reinfections are frequent, making sustained control particularly challenging. Control efforts could be strengthened through primary health care by expanding access to diagnosis and providing large-scale single-dose treatments, especially in vulnerable communities with poor sanitation. Giardiasis therefore remains an invisible and untreated disease, despite affecting millions and impairing childhood development.

Key words: *Giardia lamblia* - pathophysiology - diagnosis - treatment - health policies - giardiasis

## *Giardia lamblia*, a genetically diverse parasite, drives complex enterocyte dysfunction and growth impairment in children

*Giardia lamblia* (syn.: *Giardia duodenalis*; *Giardia intestinalis*) belongs to the phylum Metamonada, order Diplomonadida. It is a cosmopolitan gut parasitic protozoan that infects mammals and exhibits two stages: the trophozoite (the vegetative and replicative form), which measures 10-20 *μm* and features two symmetrical nuclei and four pairs of flagella; and the cyst, the infective form, which measures 8-12 *μm* and contains up to four nuclei.[1]

*Giardia lamblia* exhibits notable intraspecific genetic variability and comprises at least eight distinct assemblages, designated A through H. Assemblages A and B are the most relevant to humans, although they also occur in a range of animal hosts, suggesting zoonotic transmission.[2]

Other assemblages tend to be host-specific: C and D are primarily found in canines, E in hoofed livestock, F in cats, G in rodents, and H in marine mammals.[3] Nevertheless, reports of these assemblages in atypical hosts are not uncommon.[4] In this context, assemblage E was first unequivocally identified by our group in humans, and also in Australia.[5,6] *G. lamblia* assemblages represent deep genetic divergence and may constitute cryptic species. Sequencing of genes *gdh*, *tpi*, and *bg* allows genotyping.[7-11] The One Health approach is essential for understanding giardiasis as a complex zoonosis, where transmission among humans, domestic animals, and wildlife occurs through interconnected environmental determinants.

*Giardia lamblia* interacts intimately with the intestinal epithelium, leading to a range of inflammatory and immunological responses. It induces epithelial dysfunction through mechanisms such as disruption of tight junctions, increased intestinal permeability, and apoptosis.[12,13,14] These effects are partly mediated by the release of parasite-derived proteases and metabolic products that trigger host immune responses. Affected enterocytes produce pro-inflammatory cytokines, including interleukin-6 (IL-6), IL-8, and tumour necrosis factor (TNF), which contribute to the recruitment of immune cells and local inflammation.[15,16] Direct enterocyte damage was suggested by the detection of elevated plasma levels of intestinal fatty acid-binding protein (I-FABP), in association with increased levels of IL-17 and TNF.[17] Additionally, *G. lamblia* modulates both innate and adaptive immunity by interfering with antigen

**doi:** 10.1590/0074-02760250251

**+ Corresponding author:** filipe.anibal.carvalho.costa@gmail.com | 🔾 https://orcid.org/0000-0001-8083-2840

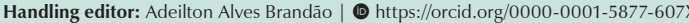

**Handling editor:** Adeilton Alves Brandão | 🔾 https://orcid.org/0000-0001-5877-607X

presentation and by inducing regulatory responses that may suppress excessive inflammation. This complex interplay between *G. lamblia* and the intestinal immune system results in malabsorption.

Between 2007 and 2014, the Bill and Melinda Gates Foundation sponsored two multicentre studies aimed at characterising the pathogens associated with two major causes of childhood mortality: acute diarrhoeal diseases and protein-calorie malnutrition. These studies were strategically commissioned within initiatives aimed at reducing mortality and malnutrition caused by enteric diseases in low- and middle-income countries. Firstly, the Global Enteric Multicentre Study (GEMS), was conducted in Kenya, Mali, Mozambique, Gambia, Bangladesh, India, and Pakistan and, using a case-control design, compared the detection rates of different enteropathogens in children with and without diarrhoea.[18] It concluded that rotavirus, *Cryptosporidium* spp., *Shigella* spp., and enterotoxigenic *Escherichia coli* are responsible for most cases and should be targeted with specific control, diagnostic, and treatment strategies. A somewhat surprising finding was that the detection rate for *G. lamblia* was consistently higher in the control group (without diarrhoea) than in the case group, which does not qualify this parasite as a cause of acute diarrhoea in children in developing countries. The unexpected nature of this finding lies in the fact that *G. lamblia* is a classic cause of acute diarrhoea in the developed world, associated with water contamination by faecal matter.[19]

The second study was the Etiology, Risk Factors, and Interactions of Enteric Infections and Malnutrition and the Consequences for Child Health and Development (the MAL-ED study), a prospective cohort study assessing the role of enteric infections in health outcomes in children followed from birth to 24 months of age. The MAL-ED study identified the chronic effects of *G. lamblia*, demonstrating that it significantly impairs children's linear growth and physical development, thus constituting a risk factor for chronic malnutrition.[20,21] Our studies reproduced these findings in the Brazilian Amazon, in the Brazilian semiarid region, and in children from urban *favelas* in Rio de Janeiro, the latter demonstrating that these effects are not related to a specific assemblage.[8,22,23] Taken together, data from the GEMS and MAL-ED studies, along with community-based cross-sectional surveys, characterise giardiasis as a chronic childhood disease that impairs intestinal absorptive function and, through a mechanism independent of diarrhoea, leads to growth and developmental delays and is a risk factor for malnutrition.

The observed impairment in growth is hypothesised to result from the combined or individual effects of environmental enteric dysfunction, malnutrition, anaemia chronic immune activation and systemic inflammation, epigenetic dysregulation, and alterations in the composition and function of the gut microbiota.[24] According to a recent review by Gutiérrez & Bartelt,[24] the available literature suggests a "triple-hit" model, in which *G. lamblia*, in association with dietary characteristics, disrupts microbiota functions and dysregulates nutrient absorption, generating potentially toxic metabolic by-products,

and restricts child growth.[25] Additionally, *G. lamblia* interacts with bile acids and digestive enzymes, impairing the digestion and absorption of fats and fat-soluble vitamins (A, D, E, and K), as well as iron and zinc micronutrients that are essential for physical and neurological development.[24]

A study conducted in Egypt reported significantly reduced serum levels of iron and zinc, along with lower body weight, in infected children compared to healthy controls.[26] This nutritional deficiency associated with giardiasis may compromise hippocampal and cerebellar development, as demonstrated by experimental evidence from an animal model: gerbils infected with *Giardia* showed tissue zinc depletion and structural alterations in the hippocampus (CA1-CA3 cortex, dentate gyrus) and cerebellum (reduced number of Purkinje cells), potentially impacting cognitive function.[27] Zinc and iron deficiencies also impair immune system function. Zinc acts as a regulator of hundreds of enzymes and plays a critical role in gene expression, cell proliferation, and differentiation particularly in immune cell lineages affecting epithelial barrier integrity and both innate and adaptive immune responses. Chronic zinc deficiency increases the secretion of pro-inflammatory cytokines, exacerbates lymphopenia and thymic atrophy, and worsens the host response to infections. In parallel, some studies have shown that giardiasis can alter both intestinal and systemic cytokine profiles through interactions between *G. lamblia* and inflammatory signalling pathways, modulating IL-8, IL-10, IL-23, and TNF, and inducing regulatory responses that may skew the Th1/Th2/Th17 balance and interfere with immune responses to infection and vaccination.[28]

## Environmental inequities as key drivers of *G. lamblia* transmission

The intersection of environmental inequalities and territorial exclusion has placed many populations at high risk of infectious diseases, including those transmitted via water and the faecal-oral route. Children living in *favelas*, impoverished rural areas without land ownership, and other settings with inadequate sanitation are particularly vulnerable. The combination of irregular water supply and inadequate faecal waste disposal facilitates the transmission of giardiasis, as has been demonstrated in several studies.[29,30,31,32,33] The Brazilian semi-arid region, for example, has historically experienced cyclical droughts. In such contexts, many communities rely on alternative water sources, such as water trucks, substantially increasing exposure to pathogenic agents.[34] In a very different rainfall scenario, regions with heavy precipitation regimes and large freshwater courses, such as the Amazon, inadequate sanitation systems in urbanised and riverine communities also facilitate the transmission of intestinal parasites such as *G. lamblia*.[9,10,20]

A third scenario — *favelas* and the outskirts of large cities — is characterised by high population density, precarious water supply, and inefficient sanitation systems that rely on transporting faecal matter via water and discharging sewage into heavily degraded watercourses.[35] These areas are criss-crossed by open ditch-

es and marked by severe environmental contamination. These landscapes highlight environmental racism, as the impacts of degradation and deterioration of living conditions fall disproportionately on historically marginalised populations mostly Black people, Indigenous peoples, and their descendants, riverine communities, rural communities in the outback of Brazil, and people living in *favelas*. In addition to facing greater socioeconomic vulnerability, these populations are excluded from decision-making processes and public policies.[36,37,38]

Children are the most affected, not only because of their greater biological susceptibility but also because of their lack of access to essential services. The ultimate outcome of environmental racism is the shaping of children's nutritional status within a multicausality framework that includes income and acute and chronic infections.[39,40] Tackling giardiasis and its determinants in Brazil therefore requires an integrated approach that goes beyond traditional treatment strategies and incorporates environmental justice actions, strengthening of primary health care, expansion of basic sanitation, and the assurance of the right to safe drinking water. It is essential to recognise that child health is directly linked to the environmental conditions in which these children live and that ignoring this reality perpetuates historical inequalities and deepens silent crises that continue to affect the most vulnerable.

## Underdiagnosis of giardiasis calls for urgent development of scalable, microscopy-free diagnostic tools

Giardiasis can be diagnosed through a variety of laboratory methods, which vary in sensitivity, specificity, and feasibility across different settings. Microscopic examination of stool samples remains a widely used, cost-effective technique and includes direct wet mounts, concentration procedures such as formalin-ethyl acetate sedimentation, and permanent staining (*e.g.*, trichrome or iodine).[41] However, the intermittent shedding of *G. lamblia* cysts and the requirement for experienced personnel limit the reliability of microscopy. To overcome these limitations, antigen detection assays targeting *Giardia*-specific coproantigens have become common in both clinical and research settings.[42]

Enzyme-linked immunosorbent assays (ELISAs) and rapid immunochromatographic tests (ICTs) have demonstrated higher sensitivity and are suitable for high-throughput screening or point-of-care use.[43] Unfortunately, these tests are still expensive and not applicable at population levels. Molecular methods, particularly polymerase chain reaction (PCR)-based assays, have provided increased sensitivity and specificity, along with the ability to genotype isolates for epidemiological studies.[44] Despite their advantages, PCR-based tools are costlier and require specialised infrastructure, which may limit their applicability in resource-limited areas. The choice of diagnostic method should be guided by the purpose of testing, available resources, and the need for accuracy in detection and characterisation of the parasite. When considered together, the low sensitivity of parasitological techniques (and their reliance

on the microscopist's expertise) and the high cost of immunological and molecular methods result in a scenario in which giardiasis is not routinely diagnosed, and most chronic infections go undetected in endemic regions.[45]

## At present, giardiasis is not addressed by any disease-specific control policy

Intestinal parasitism associated with soil-transmitted helminths [STHs (*Ascaris lumbricoides*, hookworms, and *Trichuris trichiura*)] has been a concern for the World Health Organisation (WHO) and, along with other diseases such as schistosomiasis and filariasis, has been the focus of control campaigns.[46] Despite substantial scientific evidence supporting the impact of chronic giardiasis on child growth and physical development, mass drug administration (MDA) programmes targeting intestinal parasites currently recognise only STHs and are based on the distribution of albendazole, usually as a single 400 mg dose.[47] This dosage of albendazole is ineffective against *G. lamblia*.[48,49]

The MDA strategy for preventive chemotherapy relies on drug donations from the pharmaceutical industry and philanthropic organisations, including albendazole for STHs, donated by GlaxoSmithKline. Ivermectin is donated by Merck & Co. for the control of onchocerciasis, and praziquantel is provided by Merck KGaA for MDA campaigns targeting both intestinal and urinary schistosomiasis.[50,51,52] GlaxoSmithKline also donates albendazole in combination with either ivermectin or diethylcarbamazine for the control of lymphatic filariasis. Key institutional partners supporting MDA implementation include the WHO, the Carter Centre, the Bill & Melinda Gates Foundation, United Nations International Children's Emergency Fund (UNICEF), and the World Bank, particularly in support of school health programs.[53]

The consolidation of intestinal parasite control policies without the use of parasitological diagnostics considered unfeasible because of the need for laboratory infrastructure and trained microscopists even outside of MDA, has led to the dismantling of diagnostic capacity, including specialised personnel. This, in turn, has further obscured giardiasis from the attention of philanthropic agencies, governments, and international organisations.[54,55] The development of single-dose treatments for giardiasis has also not been prioritised, and the existing alternatives, such as secnidazole and tinidazole, may lack a safety profile compatible with use in population-based MDA programmes. It is worth mentioning that indiscriminate use of imidazoles may increase the risk of emergence of drug resistant *Giardia* strains.[56,57]

## Giardiasis is a condition that is rarely adequately treated with antiparasitic drugs

The most inexpensive treatment for *G. lamblia* infection is a five-day course of oral metronidazole administered two to three times daily.[58] This treatment is usually prescribed only when laboratory confirmation is available, either through parasitological or immunological methods. As a result, most children harbouring *G. lamblia* in their gastrointestinal tract and suffering the consequences of this parasitism do not have the opportunity to

receive specific treatment and, consequently, the duration of infection throughout childhood may be prolonged.[59] Infected children also serve as sources of transmission, contaminating the environment and directly spreading *G. lamblia* in crowded settings such as day care centres.[60]

Once prescribed, completing the full course of metronidazole treatment can present practical challenges and result in poor adherence, as the drug has a bitter taste that is often rejected by children and may induce vomiting after ingestion.[61] The standard dosing regimen makes it difficult to ensure full compliance. In children, the most common side effects include nausea, abdominal pain, metallic taste, and vomiting.[62] In some settings, only tablet formulations are available, requiring crushing and improvised dilution, making it difficult to accurately adjust doses for younger children. Although treatment with metronidazole is effective in most cases, there is growing evidence of an increased frequency of refractory infections.[63,64,65] The causes of such treatment failures include inappropriate therapeutic regimens and reinfections, with an estimated 10% of cases attributed to drug-resistant parasite strains.[58,66]

Studies estimate that approximately 20% of children treated with metronidazole remain infected at follow-up cure assessments, regardless of the geographic setting.[67] Alarmingly, data from the Hospital for Tropical Diseases in London have shown an increase in treatment failures among patients returning from abroad — particularly from India — following the use of 5-nitroimidazole drugs: between 2008 and 2013, the rate of therapeutic failure rose from 15.1% to 40.2%.[68] The mechanism involved in *Giardia* drug resistance has been classically related to specific genes involved in pro-drug activation.[49] However, studies addressing *in vitro* induction of metronidazole-resistant *Giardia* showed no association with nucleotide alterations, pointing to a complex metabolic regulated process.[56] The emergence of resistance to drugs commonly used in the treatment of giardiasis is beginning to pose a threat to their continued effectiveness in clinical practice. It is important to emphasise that, in endemic areas, regardless of the underlying cause of persistent infection following treatment, prolonged exposure of the host to the parasite may occur, leading to chronic giardiasis and its associated consequences, an issue of particular concern in paediatric populations.

### Integrating giardiasis into the primary health care agenda is both feasible and necessary

Chronic giardiasis could be addressed through specific policies and control actions implemented at the primary health care level. This would require access to diagnostic tools and the distribution of single-dose drugs against *G. lamblia*. Interventions could initially prioritise vulnerable communities with low sewerage coverage. They could also focus on children within specific income brackets and those presenting low anthropometric z-scores, based on cut-off values for Z-scores (HAZ), weight-for-age Z-scores (WAZ), and body-mass-index-for-age Z-scores (BMIZ) indicators. Potential sources of contaminated water and food, as well as settings vulnerable to transmission, such as day care centres, could be mapped with the support of Community Health Workers. Primary Health Care Units could be supplied with immunological diagnostic kits and test preschool-aged children once or twice per year. Upon identification of *G. lamblia* infection, treatment with single-dose drugs or metronidazole could be administered.

Secnidazole is an oral nitroimidazole derivative effective for treating various protozoan infections, including *G. lamblia*. Compared with metronidazole, its longer half-life allows for single-dose administration, and it is also better tolerated, which increases treatment adherence. To adjust the dose in children, 30 mg per kg of body weight should be administered, which is equivalent to 1 mL per kg of body weight, in a single dose, since the concentration of the suspension is 30 mg/mL of secnidazole. More than 60 pharmaceutical companies in different countries are involved in the production of secnidazole, either as an active pharmaceutical ingredient or in finished formulations.

Most of these companies are located in India, China, and Europe, with some also in the United States. The secnidazole package insert does not state a minimum age for its use, but in general, nitroimidazoles have proven safety in children over two years of age. Tinidazole, another nitroimidazole derivative, should be administered to children over 3 years of age in a single oral dose of 50 mg/kg, and is sold as a 100 mg/mL suspension. Various pharmaceutical companies located in India, China, and in European and Latin American countries produce Tinidazole.

### Future directions and final remarks

Despite substantial advances in the understanding of *G. lamblia* biology and host-parasite interactions, the global approach to giardiasis remains fragmented and underprioritised. From a pathogenic standpoint, a more complex model of chronic intestinal dysfunction, microbiota disruption, and systemic consequences extending beyond the gut has replaced the classical notion of giardiasis as a self-limited diarrhoea disease. However, this paradigm shift has not yet translated into diagnostic, therapeutic, or public health practices. Diagnosis still relies largely on microscopy, a method that is both operator-dependent and insufficiently sensitive for detecting chronic, low-burden infections. Molecular and immunological methods could overcome these limitations, but their high cost and infrastructure requirements have restricted their use to research settings. Therapeutically, the available arsenal against *G. lamblia* remains limited, and if the indiscriminate use of antiparasitic drugs — whether driven by cultural practices or by mass treatment campaigns — continues unchecked, there is a significant risk of selecting resistant parasite isolates. This scenario is particularly concerning given the scarcity of research and development of new anti-*Giardia* compounds. From a control perspective, giardiasis remains excluded from global mass drug administration programmes and neglected on the agenda of international agencies, even though its chronic and developmental

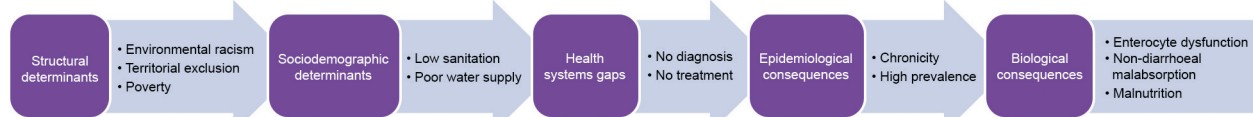

A proposal for the hierarchisation of factors involved in the genesis, development, and consequences of giardiasis in childhood.

impacts are comparable to those of other prioritised neglected tropical diseases. The convergence of scientific evidence underscores an urgent need for a paradigm shift: giardiasis should be recognised not only as a parasitic infection but also as a marker of environmental injustice and an unaddressed determinant of child health and development (Figure).

Future research should prioritise the development of point-of-care diagnostic tools, the discovery of novel therapeutic targets, and longitudinal studies integrating parasitological, immunological, nutritional, and environmental dimensions to guide evidence-based control policies. Bridging this gap requires innovation in diagnostics, safe and affordable therapies, and the inclusion of giardiasis in integrated control strategies under the framework of the Sustainable Development Goals and the One Health approach. In conclusion, a major attitudinal shift towards giardiasis is needed, involving policymakers, health authorities, and international organisations. At present, giardiasis remains an invisible and untreated condition, affecting large populations and reducing the chances of full development in a substantial proportion of children living in developing countries.

## AUTHORS' CONTRIBUTION

FACC conceived, designed and drafted the manuscript and critically revised the manuscript; AMC, DAC and MF drafted the manuscript and manuscript revision. All authors read and approved the final manuscript. The authors declare that they have no competing interests.

## DATA AVAILABILITY

The content underlying the text is included within the manuscript.

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

# OPEN PEER REVIEW

Memórias do IOC thanks the anonymous reviewers for their contribution to the peer review of this work.

**FIRST REVIEW ROUND**

**REVIEWERS' COMMENTS**

**REVIEWER #1**

Overall Assessment

This manuscript presents a comprehensive and relevant perspective on chronic Giardia lamblia infections, focusing on their pathophysiological mechanisms, socioeconomic determinants, diagnostic challenges, and lack of prioritization in global health policies. The manuscript is scientifically sound, very well organized, and written with clarity. The topic aligns perfectly with the scope of Memórias do Instituto Oswaldo Cruz, addressing parasitic diseases and public health inequities. However, same refinement to enhance accuracy, and depth this scientific work. The comments below highlight these points section by section.

Major Comments

#1. The manuscript would benefit from clearer definition of its format as a 'Perspective' paper — consider adding a short 'Future Directions' paragraph before the concluding remarks to strengthen the forward-looking element expected in this article type.

#2. Taxonomic terminology should be standardized throughout (Giardia lamblia vs. G. duodenalis vs. G. intestinalis). The consistency is essential; please select one and apply uniformly.

#3. Include recent global epidemiological data (WHO 2023–2024) in the introduction to better frame the disease burden and support the 'neglected' argument.

#4. Figure 1 requires improvement: enhance resolution and create a more explanatory caption that clearly defines the hierarchy of factors (biological, environmental, social). This reviewer believes it will be quite informative and will add clarity to the work.

#5. References 56–57 contain punctuation inconsistencies; If possible, try to include the DOI in the references; this will greatly help readers in finding the original references.; and references 58 and 66 appear to be duplicates (Gardner & Hill, 2001). Please revise and unify citation style.

Minor Comments

#1. Abstract (p.2): Slightly long; consider reducing repetition of diagnostic methods to tighten flow.

#2. Introduction (p.3): Could integrate the Savioli et al. (2006) reference earlier to link giardiasis with the Neglected Tropical Diseases Initiative.

#3. Pathophysiology section (p.4–5): Suggest clarifying mechanisms of epithelial disruption and immune modulation, citing recent reviews (2022–2024, or review 2025, if possible).

#4. Environmental Inequities section (p.7–8): Excellent sociological analysis; consider adding one figure or infographic (map) linking sanitation inequities to giardiasis prevalence in Brazil.

#5. Diagnostics section (p.9–10): Expand discussion on potential for field-deployable molecular methods (LAMP or portable PCR) and link to WHO 'Diagnostics for NTDs' 2022 roadmap.

#6. Discussion (p.11–13): Add connection between nitroimidazole resistance and antimicrobial resistance (AMR) frameworks — currently underexplored and highly relevant.

#7. Final Remarks (p.14): Strengthen policy-oriented message — e.g., integration of giardiasis screening into existing STH mass drug administration programs or other existing programs that may include the issue of giardiasis care.

Specific Page/Section Comments

p.2 – Abstract: Avoid redundancy in mentioning ELISA and ICT; briefly emphasize the 'underdiagnosis and need for policy response'.

p.3 – Line 25: Consider replacing 'major risk factor for protein-energy malnutrition' with 'significant contributor to child undernutrition'.

p.4 – Paragraph 2: When discussing Assemblages, ensure references are up to date (Thompson & Ash, 2016; Ryan & Zahedi, 2019).

p.6 – Paragraph 3: Clarify the term 'triple-hit model' by briefly defining its three components for non-specialist readers.

p.8 – Last paragraph: Excellent contextualization of environmental racism. Suggest adding a transitional sentence connecting environmental inequities to diagnostic access barriers.

p.10 – Diagnostics: Merge short paragraphs to improve cohesion and reduce redundancy regarding test sensitivity and cost.

p.11 – Drug Policy: Note that WHO's 2023 'NTD Roadmap' excludes giardiasis; mentioning this strengthens the argument for policy inclusion.

p.13 – Primary Health Care section: Consider referencing successful examples of integrating protozoan disease control into PHC, e.g., Brazilian Chagas control initiatives or malaria control.

p.14 – Final Remarks: Add a closing statement emphasizing research gaps and need for international funding mechanisms.

Recommendation

The manuscript is scientifically relevant and well written. With the suggested adjustments — mainly improve your consistency, slight expansion of contextual data, and refinement of visual elements — it will meet the high publication standards of the Mem Inst Oswaldo Cruz.

## AUTHORS' RESPONSE TO THE REVIEWERS

Dear Dr. Brandão

We thank you for considering our manuscript for publication. Below, we provide our responses to the reviewers' comments:

Reviewer: 1
Major Comments
#1. The manuscript would benefit from clearer definition of its format as a 'Perspective' paper — consider adding a short 'Future Directions' paragraph before the concluding remarks to strengthen the forward-looking element expected in this article type.

We agree and acknowledge this comment. The paragraph was added.

#2. Taxonomic terminology should be standardized throughout (Giardia lamblia vs. G. duodenalis vs. G. intestinalis). The consistency is essential; please select one and apply uniformly.

OK, we standardized Giardia lamblia.

#3. Include recent global epidemiological data (WHO 2023–2024) in the introduction to better frame the disease burden and support the 'neglected' argument.

OK, we included.

#4. Figure 1 requires improvement: enhance resolution and create a more explanatory caption that clearly defines the hierarchy of factors (biological, environmental, social). This reviewer believes it will be quite informative and will add clarity to the work.

Yes, we agree that the figure needs to be improved. We are preparing a high-resolution image for the final version. In this version, we have made adjustments to the figure text.

#5. References 56–57 contain punctuation inconsistencies; If possible, try to include the DOI in the references; this will greatly help readers in finding the original references.; and references 58 and 66 appear to be duplicates (Gardner & Hill, 2001). Please revise and unify citation style.

OK, we included.

Minor Comments
All minor comments were addressed.
#4. Environmental Inequities section (p.7–8): Excellent sociological analysis; consider adding one figure or infographic (map) linking sanitation inequities to giardiasis prevalence in Brazil.

We consider this to be an excellent suggestion; however, we intend to address this approach in a separate discussion that will also encompass additional aspects.

Specific Page/Section Comments
All comments were addressed.

## SECOND REVIEW ROUND

**REVIEWER #1**

The revised manuscript satisfactorily addresses all major and minor comments raised during the review process and now clearly conforms to the perspective article format. The inclusion of a future directions paragraph strengthens the forward-looking nature of the contribution.

The abstract is adequate and accurately reflects the scope and objectives of the manuscript. The study offers an original and relevant contribution to the field by integrating biological, environmental, and social dimensions of giardiasis, reinforcing its significance within the context of neglected diseases.

The methodology, results, and discussion are coherent and well articulated. Taxonomic standardization throughout the text improves clarity, and the inclusion of recent global epidemiological data (WHO 2023–2024) strengthens the contextualization of disease burden. The discussion is balanced and scientifically sound.

References were appropriately revised, with inconsistencies corrected, duplicate citations removed, and DOIs added where applicable. Figure 1 was improved in clarity and explanatory value, with a more informative caption and textual adjustments; a high-resolution version must be provided for final publication.

All minor comments were addressed. Overall, the authors have adequately implemented the requested revisions, and the manuscript is suitable for publication in Memórias do Instituto Oswaldo Cruz.

