## [Reviewer Report · FIRST REVIEW ROUND - REVIEWERS COMMENTS]

## REVIEWER #1

**Overall Assessment**

This manuscript presents a comprehensive and relevant perspective on chronic *Giardia lamblia* infections, focusing on their pathophysiological mechanisms, socioeconomic determinants, diagnostic challenges, and lack of prioritization in global health policies.

The manuscript is scientifically sound, very well organized, and written with clarity.

The topic aligns perfectly with the scope of Memórias do Instituto Oswaldo Cruz, addressing parasitic diseases and public health inequities.

However, same refinement to enhance accuracy, and depth this scientific work. The comments below highlight these points section by section.

**Major Comments**

#1. The manuscript would benefit from clearer definition of its format as a ‘Perspective’ paper — consider adding a short ‘Future Directions’ paragraph before the concluding remarks to strengthen the forward-looking element expected in this article type.

#2. Taxonomic terminology should be standardized throughout (*Giardia lamblia* vs. *G. duodenalis* vs. *G. intestinalis*). The consistency is essential; please select one and apply uniformly.

#3. Include recent global epidemiological data (WHO 2023–2024) in the introduction to better frame the disease burden and support the ‘neglected’ argument.

#4. Figure 1 requires improvement: enhance resolution and create a more explanatory caption that clearly defines the hierarchy of factors (biological, environmental, social). This reviewer believes it will be quite informative and will add clarity to the work.

#5. References 56–57 contain punctuation inconsistencies; If possible, try to include the DOI in the references; this will greatly help readers in finding the original references.; and references 58 and 66 appear to be duplicates (Gardner & Hill, 2001). Please revise and unify citation style.

**Minor Comments**

#1. Abstract (p.2): Slightly long; consider reducing repetition of diagnostic methods to tighten flow.

#2. Introduction (p.3): Could integrate the Savioli et al. (2006) reference earlier to link giardiasis with the Neglected Tropical Diseases Initiative.

#3. Pathophysiology section (p.4–5): Suggest clarifying mechanisms of epithelial disruption and immune modulation, citing recent reviews (2022–2024, or review 2025, if possible).

#4. Environmental Inequities section (p.7–8): Excellent sociological analysis; consider adding one figure or infographic (map) linking sanitation inequities to giardiasis prevalence in Brazil.

#5. Diagnostics section (p.9–10): Expand discussion on potential for field-deployable molecular methods (LAMP or portable PCR) and link to WHO ‘Diagnostics for NTDs’ 2022 roadmap.

#6. Discussion (p.11–13): Add connection between nitroimidazole resistance and antimicrobial resistance (AMR) frameworks — currently underexplored and highly relevant.

#7. Final Remarks (p.14): Strengthen policy-oriented message — e.g., integration of giardiasis screening into existing STH mass drug administration programs or other existing programs that may include the issue of giardiasis care.

**Specific Page/Section Comments**

p.2 – Abstract: Avoid redundancy in mentioning ELISA and ICT; briefly emphasize the ‘underdiagnosis and need for policy response’.

p.3 – Line 25: Consider replacing ‘major risk factor for protein-energy malnutrition’ with ‘significant contributor to child undernutrition’.

p.4 – Paragraph 2: When discussing Assemblages, ensure references are up to date (Thompson & Ash, 2016; Ryan & Zahedi, 2019).

p.6 – Paragraph 3: Clarify the term ‘triple-hit model’ by briefly defining its three components for non-specialist readers.

p.8 – Last paragraph: Excellent contextualization of environmental racism. Suggest adding a transitional sentence connecting environmental inequities to diagnostic access barriers.

p.10 – Diagnostics: Merge short paragraphs to improve cohesion and reduce redundancy regarding test sensitivity and cost.

p.11 – Drug Policy: Note that WHO’s 2023 ‘NTD Roadmap’ excludes giardiasis; mentioning this strengthens the argument for policy inclusion.

p.13 – Primary Health Care section: Consider referencing successful examples of integrating protozoan disease control into PHC, e.g., Brazilian Chagas control initiatives or malaria control.

p.14 – Final Remarks: Add a closing statement emphasizing research gaps and need for international funding mechanisms.

**Recommendation**

The manuscript is scientifically relevant and well written. With the suggested adjustments — mainly improve your consistency, slight expansion of contextual data, and refinement of visual elements — it will meet the high publication standards of the Mem Inst Oswaldo Cruz.

## AUTHORS’ RESPONSE TO THE REVIEWERS

Dear Dr. Brandão

We thank you for considering our manuscript for publication. Below, we provide our responses to the reviewers’ comments:

**Reviewer: 1**

**Major Comments**

#1. The manuscript would benefit from clearer definition of its format as a ‘Perspective’ paper — consider adding a short ‘Future Directions’ paragraph before the concluding remarks to strengthen the forward-looking element expected in this article type.

We agree and acknowledge this comment. The paragraph was added.

#2. Taxonomic terminology should be standardized throughout (*Giardia lamblia* vs. *G. duodenalis* vs. *G. intestinalis*). The consistency is essential; please select one and apply uniformly.

OK, we standardized *Giardia lamblia*.

#3. Include recent global epidemiological data (WHO 2023–2024) in the introduction to better frame the disease burden and support the ‘neglected’ argument.

OK, we included.

#4. Figure 1 requires improvement: enhance resolution and create a more explanatory caption that clearly defines the hierarchy of factors (biological, environmental, social). This reviewer believes it will be quite informative and will add clarity to the work.

Yes, we agree that the figure needs to be improved. We are preparing a high-resolution image for the final version. In this version, we have made adjustments to the figure text.

#5. References 56–57 contain punctuation inconsistencies; If possible, try to include the DOI in the references; this will greatly help readers in finding the original references.; and references 58 and 66 appear to be duplicates (Gardner & Hill, 2001). Please revise and unify citation style.

OK, we included.

**Minor Comments**

All minor comments were addressed.

#4. Environmental Inequities section (p.7–8): Excellent sociological analysis; consider adding one figure or infographic (map) linking sanitation inequities to giardiasis prevalence in Brazil.

We consider this to be an excellent suggestion; however, we intend to address this approach in a separate discussion that will also encompass additional aspects.

**Specific Page/Section Comments**

All comments were addressed.

---

## [Reviewer Report · REVIEWERS COMMENTS]

## REVIEWER #1

The revised manuscript satisfactorily addresses all major and minor comments raised during the review process and now clearly conforms to the perspective article format.

The inclusion of a future directions paragraph strengthens the forward-looking nature of the contribution.

The abstract is adequate and accurately reflects the scope and objectives of the manuscript.

The study offers an original and relevant contribution to the field by integrating biological, environmental, and social dimensions of giardiasis, reinforcing its significance within the context of neglected diseases.

The methodology, results, and discussion are coherent and well articulated.

Taxonomic standardization throughout the text improves clarity, and the inclusion of recent global epidemiological data (WHO 2023–2024) strengthens the contextualization of disease burden.

The discussion is balanced and scientifically sound.

References were appropriately revised, with inconsistencies corrected, duplicate citations removed, and DOIs added where applicable.

Figure 1 was improved in clarity and explanatory value, with a more informative caption and textual adjustments; a high-resolution version must be provided for final publication.

All minor comments were addressed. Overall, the authors have adequately implemented the requested revisions, and the manuscript is suitable for publication in Memórias do Instituto Oswaldo Cruz.